# Vision-Based Traffic Sign Detection and Recognition Systems: Current Trends and Challenges

**DOI:** 10.3390/s19092093

**Published:** 2019-05-06

**Authors:** Safat B. Wali, Majid A. Abdullah, Mahammad A. Hannan, Aini Hussain, Salina A. Samad, Pin J. Ker, Muhamad Bin Mansor

**Affiliations:** 1Centre for Integrated Systems Engineering and Advanced Technologies, Universiti Kebangsaan Malaysia, Bangi 43600, Malaysia; safat.wali@siswa.ukm.edu.my (S.B.W.); draini@ukm.edu.my (A.H.); salinasamad@ukm.edu.my (S.A.S.); 2Institute of Power Engineering, Universiti Tenaga Nasional, Kajang 43000, Malaysia; aaamajid2@gmail.com (M.A.A.); pinjern@uniten.edu.my (P.J.K.); muhamadm@uniten.edu.my (M.B.M.)

**Keywords:** Traffic sign detection and tracking (TSDR), advanced driver assistance system (ADAS), computer vision

## Abstract

The automatic traffic sign detection and recognition (TSDR) system is very important research in the development of advanced driver assistance systems (ADAS). Investigations on vision-based TSDR have received substantial interest in the research community, which is mainly motivated by three factors, which are detection, tracking and classification. During the last decade, a substantial number of techniques have been reported for TSDR. This paper provides a comprehensive survey on traffic sign detection, tracking and classification. The details of algorithms, methods and their specifications on detection, tracking and classification are investigated and summarized in the tables along with the corresponding key references. A comparative study on each section has been provided to evaluate the TSDR data, performance metrics and their availability. Current issues and challenges of the existing technologies are illustrated with brief suggestions and a discussion on the progress of driver assistance system research in the future. This review will hopefully lead to increasing efforts towards the development of future vision-based TSDR system.

## 1. Introduction

In all countries of the world, the important information about the road limitation and condition is presented to drivers as visual signals, such as traffic signs and traffic lanes. Traffic signs are an important part of road infrastructure to provide information about the current state of the road, restrictions, prohibitions, warnings, and other helpful information for navigation [1,2]. This information is encoded in the traffic signs visual traits: Shape, color and pictogram [1]. Disregarding or failing to notice these traffic signs may directly or indirectly contribute to a traffic accident. However, in adverse traffic conditions, the driver may accidentally or deliberately not notice traffic signs [3]. In these circumstances, if there is an automatic detection and recognition system for traffic signs, it can compensate for a driver’s possible inattention, decreasing a driver’s tiredness by helping him follow the traffic sign, and thus, making driving safer and easier. Traffic sign detection and recognition (TSDR) is an important application in the more recent technology referred to as advanced driver assistance systems (ADAS) [4], which is designed to provide drivers with vital information that would be difficult or impossible to come by through any other means [5]. The TSDR system has received an increasing interest in recent years due to its potential use in various applications. Some of these applications have been well defined and summarized in [6] as checking the presence and condition of the signs on highways, sign inventory in towns and cities, re-localization of autonomous vehicles; as well as its use in the application relevant to this research, as a driver support system. However, a number of challenges remain for a successful TSDR systems; as the performance of these systems is greatly affected by the surrounding conditions that affect road signs visibility [4]. Circumstances that affect road signs visibility are either temporal because of illumination factors and bad weather conditions or permanent because of vandalism and bad postage of signs [7]. Figure 1 shows an example of some non-ideal invariant traffic signs. These non-identical traffic signs cause difficulties for TSDR.

This paper provides a comprehensive survey on traffic sign detection, tracking and classification. The details of algorithms, methods and their specifications on detection, tracking and classification are investigated and summarized in the tables along with the corresponding key references. A comparative study on each section has been provided to evaluate the TSDR methods, performance metrics and their availability. Current issues and challenges of the existing technologies are illustrated with brief suggestions and a discussion on the progress of driver assistance system research in the future. The rest of this paper is organized as follows: In Section 2, an overview on traffic signs and recent trends of the research in this field is presented. This is followed by providing a brief review on the available traffic sign databases in Section 3. The methods of detection, tracking, and classification are categorized, reviewed, and compared in Section 4. Section 5 revises current issues and challenges facing the researchers in TSDR. Section 5 summarizes the paper, draws the conclusion and suggestions.

## 2. Traffic Signs and Research Trends

Aiming at standardizing traffic signs across different countries, an international treaty, commonly known as the Vienna Convention on Road Signs and Signals [8], was agreed upon in 1968. To date, 52 countries have signed this treaty, among which 31 are in Europe. The Vienna convention classified the traffic signs into eight categories, designated with letters A–H: Danger/warning signs (A), priority signs (B), prohibitory or restrictive signs (C), mandatory signs (D), special regulation signs (E), information, facilities or service signs (F), direction, position or indication signs (G), and additional panels (H). Examples of traffic signs in the United Kingdom for each of the categories are shown in Figure 2.

Despite the well-defined laws in the Vienna Treaty, variations in traffic sign designs still exist among the countries’ signatories to the treaty, and in some cases considerable variation within traffic sign designs can exist within the nation itself. These variations are easier to be detected by humans, nevertheless, they may pose a major challenge to an automatic detection system. As an example, different designs of stop signs in different countries are shown in Table 1.

In terms of research, recently there has been a growing interest in developing efficient and reliable TSDR systems. To show the current state of scientific research regarding this development, a simple search of the term “traffic sign detection and recognition” in the Scopus database has been carried out, with the aim of locating articles published in journals indexed in this database. To focus on the recent and most relevant research, the search has been restricted to the past decade (2009–2018) and only in the subjects of computer science and engineering. In this way, a set of 674 articles and 5414 citations were obtained. The publication and citation trends are shown in Figure 3 and Figure 4, respectively. Generally, the figures indicate a relatively fast growth rate in publications and a rapid increase in citation impact. More importantly, it is clear from the figures that TSDR research has grown remarkably in the last three years (2016–2018), with the highest number of publications and citations representing 41.69% and 60.34%, respectively.

## 3. Traffic Sign Database

A traffic sign database is an essential requirement in developing any TSDR system. It is used for training and testing the detection and recognition techniques. A traffic sign database contains a large number of traffic sign scenes and images representing samples of all available types of traffic signs: Guide, regulatory, temporary and warning signs. During the past few years, a number of research groups have worked on creating traffic sign datasets for the task of detection, recognition and tracking. Some of these datasets are publicly available for use by the research community. The detailed information regarding the publicly available databases are summarized Table 2. According to [1,9], the first and most widely used dataset is the German traffic sign dataset, which has two datasets: The German Traffic Signs Detection Benchmark (GTSDB) [10] and German Traffic Signs Recognition Benchmark (GTSRB) [11]. This dataset collects three important categories of road signs (prohibitory, danger and mandatory) from various traffic scenes. All traffic signs have been fully annotated with the rectangular regions of interest (ROIs). Examples of traffic scenes in the GTSDB database are shown in Figure 5 [12].

## 4. Traffic Sign Detection, Tracking and Classification Methods

As aforementioned, a TSDR is a driver supportive system that can be used to notify and warn the driver in adverse conditions. This system is a vision-based system that usually has the capability to detect and recognize all traffic signs, even those signs that may be partially occluded or somewhat distorted [14]. Its main tasks are locating the sign, identifying it and distinguishing one sign from another [15,16]. Thus, the procedure of the TSDR system can be divided into three stages, the detection, tracking and classification stages. Detection is concerned with locating traffic signs in the input scene images, whereas classification is about determining what type of sign the system is looking at [17,18]. In other words, traffic sign detection involves generating candidate region of interests (ROIs) that are likely to contain regions of traffic signs, while traffic sign classification gets each candidate ROI and tries to identify the exact type of sign or rejects the identified ROI as a false detection [4,19]. Detection and classification usually constitute recognition in the scientific literature. Figure 6 illustrates the main stages of the traffic sign recognition system. As indicated in the figure, the system is able to work in two modes, the training mode in which a database can be built by collecting a set of traffic signs for training and validation, and a testing mode in which the system can recognize a traffic sign which has not been seen before. In the training mode, a traffic sign image is collected by the camera and stored in the raw image database to be classified and used for training the system. The collected image is then sent to color segmentation process where all background objects and unimportant information in the image are eliminated. The generated image from this step is a binary image containing the traffic sign and any other objects similar to the color of the traffic sign. The noise and small objects in the binary image are cleaned by the object selector process and the generated image is then used to create or update the training image database. According to [20], feature selection has two functions in enhancing the performances of learning tasks. The first function is to eliminate noisy and redundant information, thus getting a better representation and facilitating the classification task. The second function is to make the subsequent computation more efficient through lowering the feature space. In the block diagram, the features are then extracted from the image and used to train the classifier in the subsequent step. In testing mode, the same procedure is followed, but the extracted features are used to directly classify the traffic sign using a pre-trained classifier.

Tracking is used by some research in order to improve the recognition performance [21]. The three stages of a TSDR system are shown in Figure 7, and further discussed in the subsequent sections.

### 4.1. Detection Phase

The initial stage in any TSDR system is locating potential sign image regions from a natural scene image input. This initial stage is called the detection stage, in which a ROI-containing traffic sign ise actually localized [17,23,24]. Traffic signs usually have a strict color scheme (red, blue, and white) and specific shapes (round, square, and triangular). These inherent characteristics distinguish them from other outdoor objects making them suitable to be processed by a computer vision system automatically, thus, allow the TSDR system to distinguish traffic signs from the background scene [21,25]. Therefore, traffic sign detection methods have been traditionally classified into color-based, shape-based and hybrid (color–shape-based) methods [23,26]. Detection methods are outlined in Figure 8 and compared in the following subsections.

#### 4.1.1. Color-Based Methods

Color-based methods take advantage of the fact that traffic signs are designed to be easily distinguished from their surroundings, often colored in highly visible contrasting colors [17]. These colors are extracted to detect ROI within an input image based on different image-processing methods. Detection methods based on the color characteristics have low computing, good robustness and other characteristics, which can improve the detection performance to a certain extent [25]. However, methods based on color information can be used with a high-resolution dataset but not with grayscale images [23]. In addition, the main problem with using the color parameter is its sensitivity to various factors such as the distance of the target, weather conditions, time of the day, as well as reflection, age and condition of the signs [17,23].

In color-based approaches, the captured images are partitioned into subsets of connected pixels that share similar color properties [26]. Then the traffic signs are extracted by color thresholding segmentation based on smart data processing. The choice of color space is important during the detection phase, hence, the captured images are usually transformed into a specific color space where the signs are more distinct [9]. According to [27], the developed color-based detection methods are based on the red, green, blue (RGB) color space [28,29,30], the hue, saturation, and value (HSV) color space [31,32], the hue, saturation, and intensity (HSI) color space [33] and various other color spaces [34,35]. The most common color-based detection methods are represented in Figure 9 and reviewed respectively in Table 3.

Color thresholding segmentation is one of the earliest techniques used to segment digital images [26]. Generally, it is based on the assumption that adjacent pixels whose value (grey level, color value, texture, etc.) lies within a certain range belong to the same class [36]. Normal color segmentation was used for traffic sign detection by Varun et al. [37] with their own created dataset, containing 2000 test images, resulting in an accuracy level of 82%. The efficiency was improved in [38] by using color segmentation followed by a color enhancement method. In recent research, color thresholding has commonly been used for pre-processing purposes [39,40]. In [39], pre-filtering was used to train a color classifier, which created a regression problem, whose core was to find a linear function, as shown in (1).
(1)$$f(x)=(w,x)+b,x={\left(v_{1},v_{2},v_{3}\right)}^{i}$$
where $v_{i}$ is the intensity value of $i_{th}$ channel (i = 1, 2, 3 for a three-channel RGB image), (w,b)∈ℜ×ℜ are parameters that control the function and the decision rule is given by sgn(f(x)). In [40], Vazquez-Reina et al. used RGB to HSI color space conversion with the additional feature of white sign detection. The main advantage of this feature is its illuminated sign detection. In Refs. [33,41,42,43,44,45], HSI/HSV transformation approach was used for the purpose of detection. The major advantages of the HSI color space over the RGB color space are that it has only two components, hue and saturation, both are very similar to human perception and it is more immune to lighting conditions. In [33], a simple RGB to HSI color space transformation is used for the TSDR purpose. In [44], the HSI color space was used for detection, and then, the detected signal was passed to the distance to borders (DtBs) feature for shape detection to increase the accuracy level. The average accuracy was approximately 88.4% on GRAM database. The main limitation of using HSV transformation is the strong hue dependency of brightness. Hue is only a measurement of the physical lightness of a color, not the perceived brightness. Thus, the value of a fully saturated yellow and blue is the same.

Region growing is another simple and popular technique used for detection in TSDR systems. Region growing is a pixel-based image segmentation method that starts by selecting a starting point or seed pixel. Then, the region develops by adding neighboring pixels that are uniform, according to a certain match criterion, increasing step-by-step the size of the region [46]. This method was used by Nicchiotti et al. [47] and Priese et al. [48] for TSDR. Its efficiency was not very high, approximately 84%. Because this method is dependent on seed values, problems can occur when the seed points lie on edges, and, if the growth process is dominated by the regions, uncertainty around edges of adjacent regions may not be resolved correctly.

The color indexing method is another simple method that identifies objects entirely on the basis of color [49]. It was developed by Swain and Ballard [50] and was used by researchers in the early 1990s. In this method, a comparison of any two colored images is done by comparing their color histogram. For a given pair of histograms, *I* and *M*, each containing *n* bins, the histogram intersections are defined as [50]:(2)$$\sum_{j=1}^{n}\min\left(I_{j},M_{j}\right).$$

The match value is then,
(3)$$H(I,M)=\frac{\sum_{j=1}^{n}\min\left(I_{j},M_{j}\right)}{\sum_{j=1}^{n}M_{j}}.$$

The advantage of using color histograms is their robustness with respect to geometric changes of projected objects [51]. However, color indexing is segmentation dependent, and complete, efficient and reliable segmentation cannot be performed prior to recognition. Thus, color indexing is negatively characterized as being an unreliable method.

Another approach to color segmentation is called a dynamic pixel aggregation [52]. In this method, the segmentation process is accomplished by introducing a dynamic thresholding to the pixel aggregation process in the HSV color space. The applied threshold is independent in terms of linearity and its value is defined as [52],
(4)$$a=k-\sin(s_{seed})$$
where, *k* is the normalization parameter and *S_seed_* is the seed pixel saturation. The main advantage of this approach is hue instability reduction. However, it fails to reduce other segmentation-based problems, such as fading and illumination. This method was tested in [52] on their own created database with 620 outdoor images, resulting in an accuracy level approximately 86.3 to 95.7%.

The International Commission on Illumination 1997 Interim Color Appearance Model (CIECAM97) appearance model is another method has been used to detect and extract color information and to segment and classify traffic signs. Generally, color appearance models are capable of predicting color appearance under a variety of viewing conditions, including different light sources, luminance levels, surrounds, and lightness of backgrounds [53]. This model was used by Gao et al. [54] to transform the image from RGB to (International Commission on Illumination) CIE XYZ values. The main drawback of this model is its chromatic-adaptation transform, which is called the Bradford transform, where chromatic blues appear purple as the chroma is reduced at a constant hue angle.

The Green (Y), Blue (Cb), Red (Cr) (YCbCr) color space has been considered in recent approaches. Different from the most common color space RGB, which represents color as red, green and blue, YCbCr represents color as brightness and two-color difference signals. It was used for detection in [55], showing an accuracy level over 93% on their own collected database. The efficiency was improved to approximately 97.6% in [56] by first transforming RGB color space to YCbCr color space, then segmenting the image and performing shape-based analysis.

#### 4.1.2. Shape-Based Methods

Just as traffic signs have specific colors, they also have very well-defined shapes that can be searched for. Shape-based methods ignore the color in favor of the characteristic shape of signs [17]. Detection of a traffic sign via its shape follows the defining algorithm of shape detection i.e., to finding the contours and approximating it to reach a final decision based on the number of contours [15,23]. Shape detection is preferred for traffic signs recognition as the colors found on traffic signs changes according to illumination. In addition, shape detection reduces the search for a road sign regions from the whole image to a small number of pixels [57]. However, for this method the memory and computational requirement is quite high for large images [58]. In addition, damaged, partially obscured, faded and blurred traffic signs may cause difficulties in detecting traffic signs accurately, leading to a low accuracy rate. Detection of the traffic signs in these methods is made from the edges of the image analyzed by structural or comprehensive approaches [23]. Many shape-based methods are popular in TSDR systems. These methods are represented in Figure 10 and reviewed respectively in Table 4.

The most common shape-based approach is the Hough transformation. The Hough transformation usually isolates features of a particular shape within a given frame/images [15]. It was applied by Zaklouta et al. in [59] to detect triangular and circular signs. Their own test datasets contained 14,763 and 1584 signs, and the accuracy rate was approximately 90%. The main advantage of the Hough transformation technique is that it is tolerant of gaps in feature boundary descriptions and is relatively unaffected by image noise [60]. However, its main disadvantage is the dependency on input data. In addition, it is only efficient for a high number of votes that fall in the correct bin. When the parameters are large, the average number of votes cast for a single bin becomes low, and thus, the detection rate is decreased.

Another shape-based detection method is the similarity detection. In this method the detection is performed by computing a similarity factor between a segmented region and set of binary image samples representing each road sign shape [57]. This method was used by Vitabile et al. [52] on their collected dataset with an accuracy level over 86.3%. The main advantage of this method is its straightforwardness, whilst its main drawback is that the input image should be perfectly segmented and the dimensions have to be same. In [52], the images were initially converted from RGB to HSV, then they were segmented and resized into 36 × 36 pixels. The similarity detection equation is,
(5)$$x^{\prime }=\frac{x-x_{\min}}{x_{\max}-x_{\min}}.ny^{\prime }=\frac{y-y_{\min}}{y_{\max}-y_{\min}}.n$$
where, $x_{\max}$, $y_{\max}$, $x_{\min}$ and $y_{\min}$ are the coordinates of the rectangle vertices.

Distance transform matching (DTM) is also another type of shape-based detection method. In this method, the distance transform of the image is formed by assigning each non-edge pixel a value that is a measure of distance to the nearest edge pixel. It was used by Gavrila [61] to capture large variations in object shape by identifying the template features to the nearest feature image from a distribution of distances. This distance is inversely proportional to the matching of the image and the templates of the images. The chamfer distance equation is:(6)$$D_{chamfer}(T,I)\equiv \frac{1}{T}\sum_{t\in T}d_{I}(t)$$
where |T| and $d_{I}(t)$ denote the number of features and the distance between features t in T and the closest feature in I, respectively. In his experiment, Gavrila [61] used DTM to examine 1000 collected test images, and the accuracy was approximately 95%. The DTM technique is efficient for detecting arbitrary shapes within images. However, its main disadvantage is the vulnerability of detecting cluttered images.

Another popular two colorless traffic sign detection methods are edge detection features and Haar-like features. Edge detection refers to the process of identifying and locating sharp discontinuities in an image [62]. By using this method, image data is simplified for the purpose of minimizing the amount of data to be processed. This method was used in [63,64,65,66,67] for indicating the boundaries of objects within the image through finding a set of connected curves. The Haar-like features method was proposed by Paul Viola and Michael Jones [68] based on the Haar wavelet to recognize the target objects. As indicated in Table 4, the Haar-like features based detection method was used in [69,70] for traffic sign detection. The main advantage is its calculating speed, where any size of images can be calculated in a constant time. However, its weakness is the requirement of a large number of training images and high false positive rates [23].

#### 4.1.3. Hybrid Methods

As previously discussed, both color-based and shape-based methods have some advantages and disadvantages. Therefore, researchers recently have tried to improve the efficiency of the TSDR system using a combination of color- and shape-based features. In the hybrid methods, either color-based approaches take shape into account after having looked at colors, or shape detection is used as the main method but integrate some color aspects as well. In color-based approaches a two-stage strategy is usually employed. First, segmentation is done to narrow the search space. Subsequently, shape detection is implemented and is applied only to the segmented regions [58]. Color and shape features were combined into traffic sign detection algorithms in studies [71,72,73,74,75,76]. In these studies, different signs with various colors and shapes were detected using different datasets.

### 4.2. Tracking Phase

For robust detection and in order to increase the accuracy of the information used in identifying traffic signs, the signs are tracked using a simple motion model and temporal information propagation. This tracking process is very important for real-time applications, by which the TSDR system verifies correctness of the traffic sign and keeps tracking the sign to avoid handling the same detected sign more than once [21,83]. The tracking process is performed by feeding the TSDR system with a video recorded by a camera fixed on the vehicle and monitoring the sign candidates on a number of consecutive frames. The accepted sign candidates are only those shown up more than once. If the object is not a traffic sign or a sign that only shows up once, it can be eliminated as soon as possible, and thus, the computation time of the detection task can be reduced [84]. According to [85] and as shown in Table 5, the most common tracker adapted is the Kalman filter, as in [82,85,86,87,88]. The block diagram of a TSDR system with a tracking process based on the Kalman filter as proposed in [82] is shown in Figure 11. In the figure, SIFT, CCD and MLP are abbreviations of scale-invariant feature transform, contracting curve density and multi-layer perceptrons, respectively.

### 4.3. Classification Phase

After the localization of ROIs, classification techniques are employed to determine the content of the detected traffic signs [1]. Understanding the traffic rule enforced by the sign is achieved by reading the inner part of the detected traffic sign using a classifier method. Classification algorithms are neither color-based nor shape-based. The classifier usually takes a certain set of features as the input, which distinguishes the candidates from each other. Different algorithms are used to classify the traffic signs swiftly and accurately. Some conventional methods used for classification of traffic signs are outlined in Figure 12 and reviewed respectively in Table 6, Table 7, Table 8, Table 9, Table 10, Table 11, Table 12 and Table 13.

Template matching is a common method in image processing and pattern recognition. It is a low-level approach which uses pre-defined templates to search the whole image pixel by pixel or to perform the small window matching [15]. It was used for TSDR by Ohara et al. [90] and Torresen et al. [91]. It has the advantages of being fast, straightforward and accurate (with a hit rate of approximately 90% on their own pictured images dataset). However, the drawback of this method is that it is very sensitive to noise and occlusions. In addition, it requires a separate template for each scale and orientation. Examples of TSDR systems using a template matching method are shown in Table 6.

Another common classification method is the random forest. It is a machine learning method that operates by constructing a multitude of decision trees during the training time and outputting the class that is the mode of the output of the class of individual trees. This method was compared in [92,93] with SVM, MLP, Histogram of Oriented Gradient (HOG)-based classifiers, showing the highest accuracy rate and the lowest computational time. Based on their own dataset, the accuracy was approximately 94.2%, whereas the accuracy of the SVM is 87.8% and that of MLP is 89.2%. In terms of computational time for a single classification, the SVM takes 115.87 ms, MLP takes 1.45 ms, and a decision tree takes 0.15 ms. Despite its high accuracy and low computation time, the main limitation of a random forest is that a large number of trees can make the algorithm slow and ineffective for real-time predictions. Examples of TSDR systems using a decision tree method are shown in Table 7.

Genetic algorithm is another classification method. It is based on a natural selection process that mimics biological evolution, which was used early in this century. This method was used for traffic sign recognition by Aoyagi et al. [98] and Eccalera et al. [99]. It was proved in these studies that this method is effective in detection of the traffic sign even if the traffic sign has some shape loss or illumination problem. The disadvantage of the genetic algorithm is non-deterministic work time and non-guarantee finding of the best solution [57]. Examples of TSDR systems using a genetic algorithm method are shown in Table 8.

The other most common method for classification is using an artificial neural network (ANN). This method has gained an increasing popularity in recent years due to the advancement in general-purpose computing on graphics processing units (GPGPU) technologies [2]. In addition, it is popular due to its robustness, greater adaptability to changes, flexibility and high accuracy rate [100]. Another key advantage of this method is its ability to recognize and classify objects at the same time, while maintaining high speed and accuracy [2]. ANN-based classifiers were used in [56,99,101,102,103,104,105,106,107,108] for TSDR. In the experiment conducted in [56], the hit rate was 97.6%, and the computational time was 0.2 s. However, in [107] ANN-based methods were described to have some limitations, such as their slowness and the instability in the NN training due to too large a step. This method was compared with a template matching method in [108], concluding that NNs require a large number of training samples for real world applications. Examples of TSDR systems using an ANN method are shown Table 9.

Another increasingly popular method in vision-based object recognition is the deep learning method. This method has acquired general interest in recent years owing to its high performance of classification and the power of representational learning from raw data [109,110]. Deep learning is a part of a broader family of machine learning methods. In contrary to task specific methods, deep learning focuses on data representations with supervised, weakly supervised or unsupervised learning. Deep learning methods use a cascade of many layers of nonlinear processing units for feature extraction and transformation. Each successive layer uses the output from the previous layer as input. Higher level features are derived from lower level features to form a hierarchical representation [110]. Among the deep learning models, the convolutional neural networks (CNN) have acquired unique noteworthiness from their repeatedly confirmed superiorities [111]. According to [112], CNN models are the most widely used deep learning algorithms for traffic sign classification to date. Of the examples applied to traffic sign classification are committee CNN [113], multi-scale CNN [114], multi-column CNN [102], multi-task CNN [111,115], hinge-loss CNN [116], deep CNN [46,117], a CNN with diluted convolutions [118], a CNN with a generative adversarial network (GAN) [119], and a CNN with SVM [120]. Based on these studies, a simultaneous detection and classification can be achieved using deep learning-based methods. This simultaneousness results in improved performance, boosted training and testing speeds. Examples of TSDR systems using a deep learning method are shown in Table 10.

Adaptive boosting or AdaBoost is a combination of multiple learning algorithms that can be utilized for regression or classification [15]. It is a cascade algorithm, which was introduced by Freund and R. Schapire [122]. Its working concept is based on constructing multiple weak classifiers and assembling them into a single strong classifier for the overall task. As indicated in Table 11, the AdaBoost method was used for TSDR in [123,124,125,126,127]. Based on these studies, it can be concluded that the main advantage of the AdaBoost is its simplicity, high prediction power and capability to cascade an architecture for improving the computational efficiency. However, its main disadvantage is that if the input data have wide variations or abrupt changes in the background, then the training time increases and classifier accuracy decreases [121]. In addition, the AdaBoost trained classifier cannot be dynamically adjusted with new coming samples unless retrained from the beginning, which is time consuming and demands storing all historical samples [128]. Examples of TSDR systems using an AdaBoost method are shown in Table 11.

Support vector machine (SVM) is another classification method that contracts an N-dimensional hyper plane that optimally separates the data into two categories. More precisely, SVM is a binary classifier that separates two different classes by a subset of data samples called support vectors. It was implemented as a classifier for traffic sign recognition in [44,55,88,129,130,131,132,133,134,135,136]. This classification method is robust, highly accurate and extremely fast which is a good choice for large amounts of training data. In [129], a SVM-based classifier was applied for detecting speed limit signs and it was compared with the artificial neural network multilayer perceptron (MLP), k-nearest neighbors (kNN), least mean squares (LMS), least squares (LS) and extreme learning machine (ELM) based classifiers. Results of the comparison demonstrated that the SVM-based classifier obtained the highest accuracy and lowest standard deviation amongst all other classifiers. Similarly, in a recent study [3], a cascaded linear SVM classifier was used for detecting speed limit signs, and the result was a recall of 99.81% with a precision of 99.08% on the GTSRB dataset. In [55], a SVM-based classifier was used to detect and classify red road signs in 1000 test images, and the accuracy rate was over 95%. In [88,131], SVM was used with Gaussian kernels for the recognition of traffic signs, and the success rate was 92.3% and 92.6%, respectively. In [136], an advanced SVM method was proposed and tested with binary pictogram and gray scale images; the result was achieving high accuracy rates of approximately 99.2% and 95.9%, respectively. SVM has also shown great effectiveness in extracting the most relevant shots of an event of interest in a video, where a new SVM-based classifier called nearly-isotonic SVM classifier (NI-SVM) was proposed in [137] for prioritizing the video shots using a novel notion of semantic saliency. The proposed classifier exhibited higher discriminative power in event analysis tasks. The main disadvantage of SVM is lack of transparency of results. Transparency means how the results were obtained by the kernel and how the results should be interpreted. In SVM such things are unknown and cannot be known due to the high dimensional vector space. Examples of TSDR systems using a SVM method are shown in Table 12.

In addition to these conventional methods, researchers have used other methods for recognition. In [138], the SIFT matching method was used for recognizing broken areas of a traffic sign. This method adjusts the traffic sign to a standard camera axis and then compares it with a reference image. Sebanja et al. in [139] used principal component analysis (PCA) for both TSDR and the accuracy rate was approximately 99.2%. In [140], the researchers used improved fast radial symmetry (IFRS) for detection and a pictogram distribution histogram (PDH) for recognition. Soheilian et al. in [141] used template matching followed by a three dimensional (3D) reconstruction algorithm to reconstruct the traffic signs obtained from video data and to improve the visual angle for detecting traffic signs. In [142], Pei et al. used low rank matrix recovery (LRMR) to recover the correlation for classification with a hit rate of 97.51% in less than 0.2 s. Gonzalez-Reyna et al. [143] used oriented gradient maps for feature extraction, which is invariant in illumination and variable lighting. For classification, they used Karhunen–Loeve transform and MLP. They reported an accuracy of 95.9% and processing time of 0.0054 s per image. In [35], Miguel et al. used a self-organizing map (SOM) for recognition, where in every level, a pre-processor extracts a feature vector characterizing the ROI and passes it to the SOM. The accuracy rate was very high, approximately 99%. Examples of TSDR systems using the other methods are shown in Table 13.

## 5. Current Issues and Challenges

TSDR is the essential part of the ADAS. It is mainly designed to operate in a real-time environment for enhancing driver safety through the fast acquisition and interpretation of traffic signs. However, there are a number of external non-technical challenges that may face this system in the real environment degrading its performance significantly. Among the many issues that needed to be addressed while developing a TSDR system are the following issues outlined in Figure 13.

Variable lighting condition: Variable lighting condition is one of the key issues to be considered during TSDR system development. As aforementioned, one of the main distinguishing features of traffic sign is its unique colors which discriminate it from the background information, thus facilitating its detection. However, in outdoor environments illumination changes greatly affects the color of traffic sign, making the color information become completely unreliable as a main feature for traffic sign detection. To cope with such challenge, a method based on adaptive color threshold segmentation and high efficient shape symmetry algorithms has been recently proposed by Xu et al. [26]. This method is claimed to be robust for a complex illumination environment, exceeding a detection rate of 94% on GTSDB dataset.

Fading and blurring effect: Another important difficulty for a TSDR system is the fading and blurring of traffic signs caused by illumination through rain or snow. These conditions can lead to increase in false detections and reduce the effectiveness of a TSDR system. Using a hybrid shape-based detection and recognition method in such conditions can be very useful and may give more superior performance [146].

Affected visibility: Light emitted by the headlamps of the incoming vehicles, shadows, and other weather-related factors such as rains, clouds, snow and fog can lead to poor visibility. Recognizing traffic signs from a road image taken in such cases is a challenging task, and a simple detector may fail to detect these traffic signs. To resolve this problem, it is necessary to enhance the quality of taken images and make them clear by using an image pre-processing technique. A pre-processing makes image filtration and converts input information into usable format for further analysis and detection [147].

Multiple appearances of sign: While detecting traffic signs mainly in city areas, which are more crowded by signs, multiple traffic sign appearing at a time and similar shape man-made objects can cause overlapping of signs and lead to a false detection. The detection process can also be affected by rotation, translation, scaling and partial occlusion. Li et al. in [33], used HSI transform and fuzzy shape recognizer which is robust and unaffected by these problems and its accuracy rate in different weather condition is; sunny 94.66%, cloudy 92.05%, rainy 90.72%.

Motion artifacts: In the ADAS application, the images are captured from a moving vehicle and sometimes using a low resolution camera, thus, these images usually appear blurry. Recognition of blurred images is a challenging task and may lead to false results. In this respect, a TSDR system that integrates color, shape, and motion information could be a possible solution. In such a system, the robustness of recognition is improved through incorporating the detection and classification with tracking using temporal information fusion [73]. The detected traffic signs are tracked, and individual detections from sequential frames (t−t0, …, t) are temporally fused for a robust overall recognition.

Damaged or partially obscured sign: The other distinctive feature of traffic sign is its unique shape. However, traffic signs could appear in various conditions including damaged, partly occluded and/or clustered. These conditions can be very problematic for the detection systems, particularly shape-based detection systems. In order to overcome these problems, hybrid color segmentation and shape analysis based methods are recommended [15].

Unavailability of public database: A database is a crucial requirement for developing any TSDR system. It is used for training and testing the detection and recognition methods. One of the obstacles facing this research area is the lack of large, properly organized, and free available public image databases. According to [12], for example, the most commonly used database (GTSDB database) contains only 600 training images and 300 evaluation images. Of the seven categories classified in the Vienna convention, GTSDB covers only three categories of traffic signs for detection: prohibitory, mandatory and danger. All included images are only German traffic signs, which are substantially different from other parts of the world. To resolve the database scarcity problem, perhaps one of the ideas is to create a unified global database containing a large number of images and videos for road scenes in various countries around the world. These scenes must contain all categories of traffic signs under all possible weather conditions and physical states of the signs.

Real-time application: The detection and recognition of traffic signs are caught up with the performance of a system in real-time. Accuracy and speed are surely the two main requirements in practical applications. Achieving these requirements requires a system with efficient algorithms and powerful hardware. A good choice is convolutional neural networks-based learning methods with GPGPU technologies [2].

In brief, although lots of relevant approaches have been presented in the literature, no one can solve the traffic sign recognition problem very well in conditions of different illumination, motion blur, occlusion and so on. Therefore, more effective and more robust approaches need to be developed [12].

## 6. Conclusions and Suggestion

The major objective of the paper was to analyze the main direction of the research in the field of automatic TSDR and to categorize the main approaches into particular sections to make the topics easy to understand and to visualize the overall research for future directions. Unlike most of the available review papers, the scope of this paper has been broadened to cover all recognition phases: Detection, tracking and classification. In addition, this paper has tried to discuss as many studies as possible, in an attempt to provide a comprehensive review of the various alternative methods available for traffic sign detection and recognition; including along with methods categorization, current trends and research challenges associated with TSDR systems. The overall summary is presented in Figure 14.

The conducted review reveals that research in traffic sign detection and recognition has grown rapidly, where the number of papers published during the last three years was approximately 280 papers, which represents about 41.69% of the total number of papers published during the last decade as a whole. With regard to the methods used, it was observed that the subject of traffic sign detection and recognition incorporates three main steps: Detection, tracking and classification; and in each step, many methods and algorithms were applied, each has its own merits and demerits. In general, the methods applied in detection and recognition consider either color or shape information of the traffic sign. However, it is well known that the image quality in real-world traffic scenarios is usually poor; due to low resolution, weather condition, varying lighting, motion blur, occlusion, scale and rotation and so on. In addition, traffic signs are usually in a variety of appearances, with high inter-class similarity, and complicated backgrounds. Thus, proper integration of color and shape information in both detection and classification phases is a very promising and exciting task that is in need of much more attention. For tracking, the Kalman filter and its variations are the most common methods. For classification, artificial neural network and support vector machine-based methods were found to be the most popular methods, with a high detection rate, high flexibility and easy adoptability. Despite the recent improvements in the overall performance of TSDR systems, more research is still needed to achieve a rigorous, robust and reliable TSDR system. It is believed that TSDR system performance can be enhanced by merging the detection and classification tasks into one step rather than performing them separately. By doing so, classification can improve detection and vice versa. Another idea for further improvement of TSDR is by using standard, sufficient and large databases for learning, testing and evaluation of the proposed algorithms. In this way, the TSDR system will be able to recognize the eight different categories of the traffic signs in the real environment with different conditions. This paper will be a useful reference for researchers looking for an understanding of the current status of research in the field of TSDR and finding the related research problems in need of solutions.

## Figures and Tables

**Figure 1 sensors-19-02093-f001:**
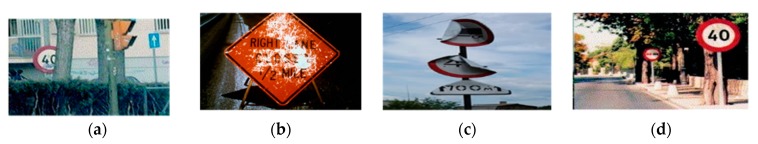
Non-identical traffic signs: (**a**) Partially occluded traffic sign, (**b**) faded traffic sign, (**c**) damaged traffic sign, (**d**) multiple traffic signs appearing at a time.

**Figure 2 sensors-19-02093-f002:**
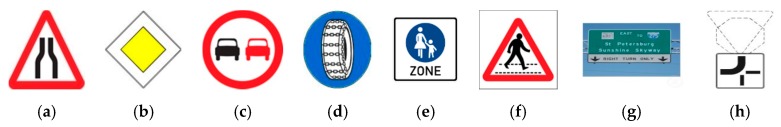
Examples of traffic signs: (**a**) A danger warning sign, (**b**) a priority sign, (**c**) a prohibitory sign, (**d**) a mandatory sign, (**e**) a special regulation sign, (**f**) an information sign, (**g**) a direction sign and (**h**) an additional panel.

**Figure 3 sensors-19-02093-f003:**
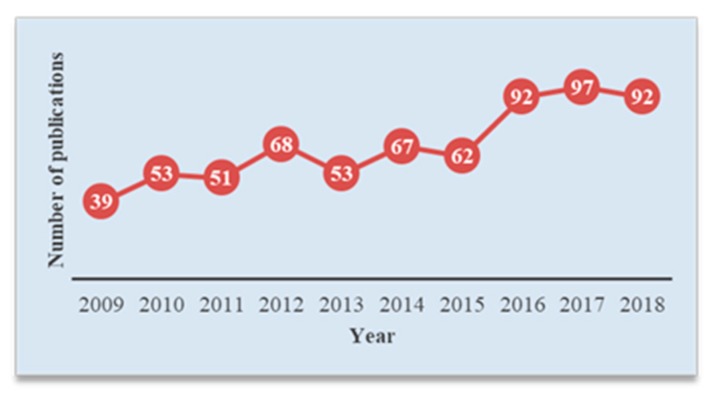
Trends of research for a traffic sign detection and recognition (TSDR) topic based on Scopus analysis tools.

**Figure 4 sensors-19-02093-f004:**
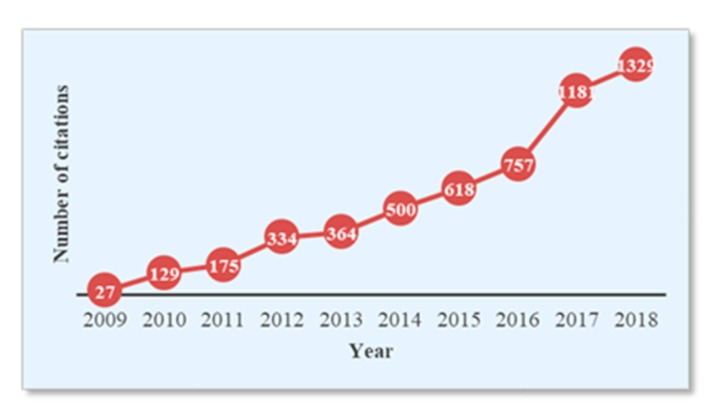
Trends of citations for a TSDR topic based on Scopus analysis tools.

**Figure 5 sensors-19-02093-f005:**
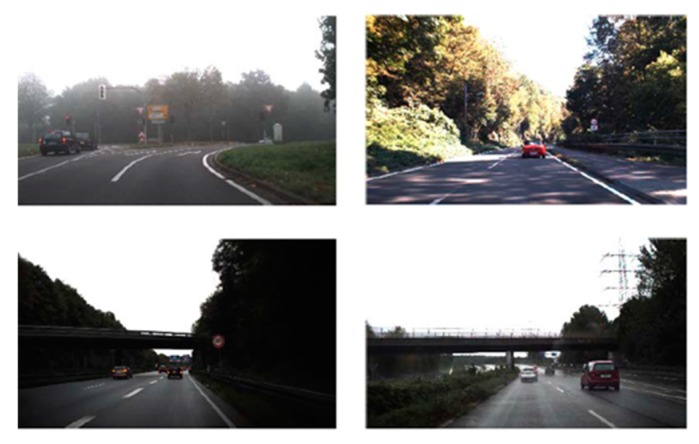
Examples of traffic scenes in the German Traffic Signs Detection Benchmark (GTSDB) database [12].

**Figure 6 sensors-19-02093-f006:**
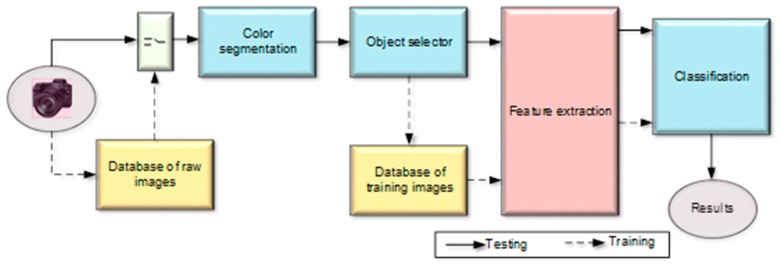
Block diagram of the traffic sign recognition system.

**Figure 7 sensors-19-02093-f007:**
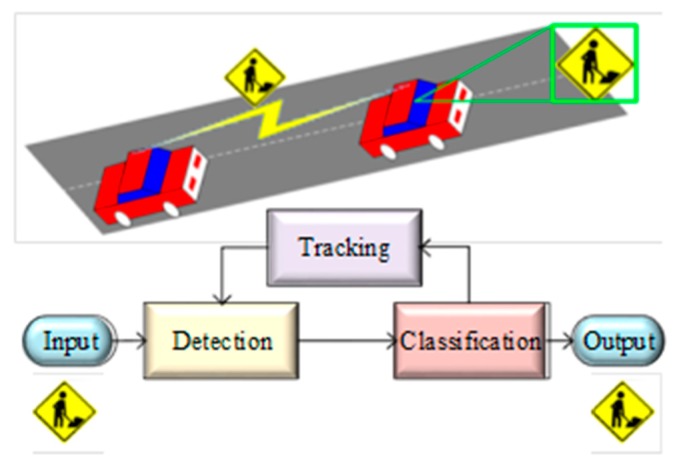
General procedure of TSDR system [22].

**Figure 8 sensors-19-02093-f008:**
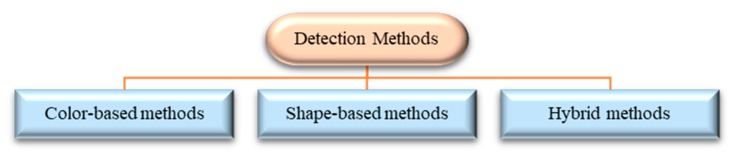
Different methods applied for traffic sign detection.

**Figure 9 sensors-19-02093-f009:**
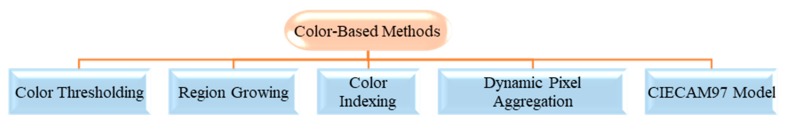
Most popular color-based detection methods.

**Figure 10 sensors-19-02093-f010:**
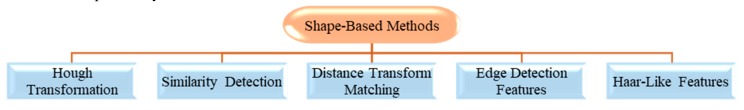
Most popular shape-based detection methods.

**Figure 11 sensors-19-02093-f011:**
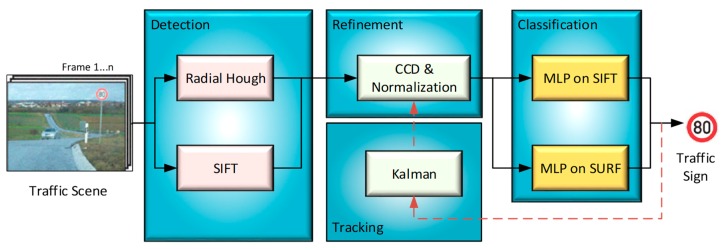
An example of a TSDR system includes tracking process based on Kalman filter [81].

**Figure 12 sensors-19-02093-f012:**
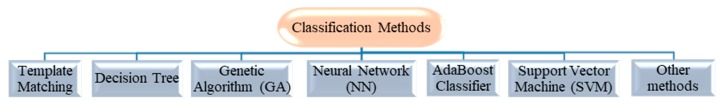
Most popular classification methods.

**Figure 13 sensors-19-02093-f013:**
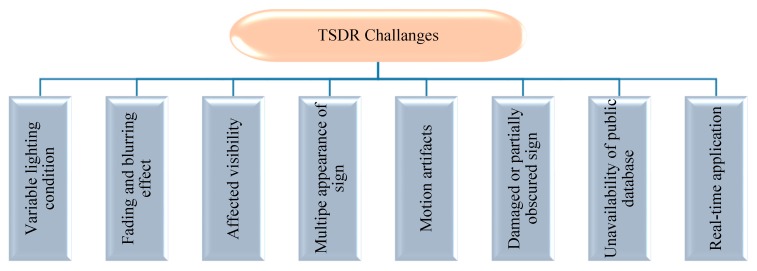
Some of TSDR challenges.

**Figure 14 sensors-19-02093-f014:**
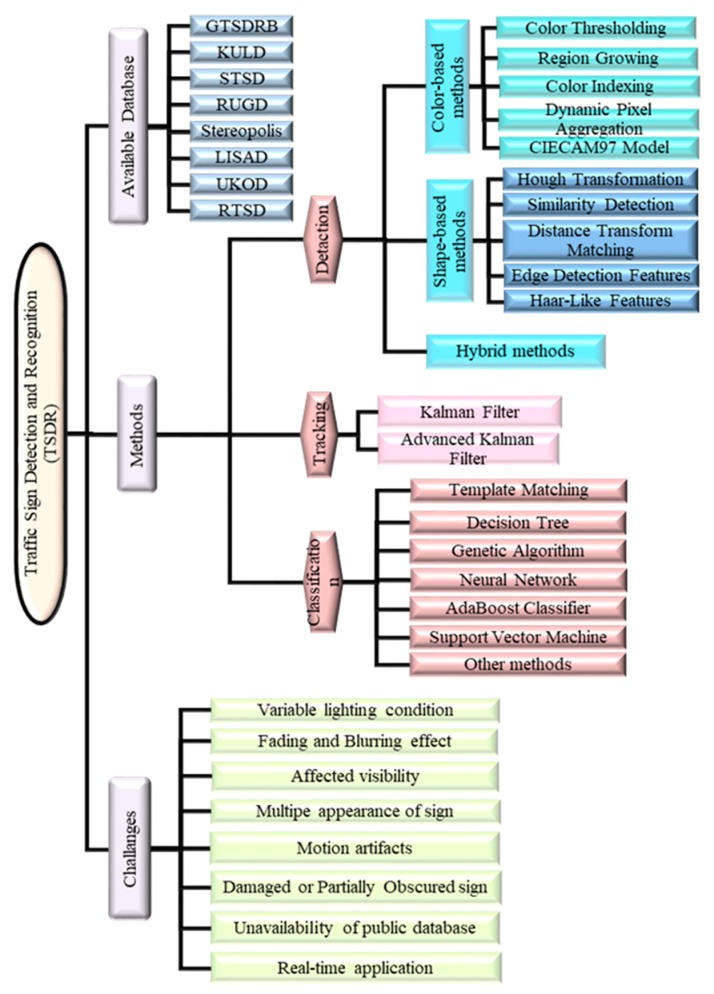
Summary of the paper.

**Table 1 sensors-19-02093-t001:** Example of stop signs in different countries.

Country	US	Japan	Pakistan	Ethiopia	Libya	New Guinea
**Sign**	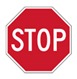	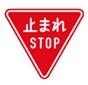	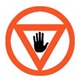	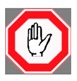	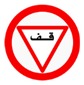	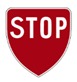

**Table 2 sensors-19-02093-t002:** Publicly available traffic sign databases [13].

Dataset	Country	Classes	TS Scenes	TS Images	Image Size (px)	Sign Size (px)	Include Videos
**GTSDRB** **(2012 and 2013)**	Germany	43	9000	39,209 (training), 12,630 (testing)	15 × 15 to 250 × 250	15 × 15 to 250 × 250	No
**KULD** **(2009)**	Belgium	100+	9006	13,444	1628 × 1236	100 × 100 to 1628 × 1236	Yes, 4 tracks
**STSD** **(2011)**	Sweden	7	20,000	3488	1280 × 960	3 × 5 to 263 × 248	No
**RUGD** **(2003)**	The Netherlands	3	48	48	360 × 270	N/A	No
**Stereopolis** **(2010)**	France	10	847	251	1920 × 1080	25 × 25 to 204 × 159	No
**LISAD** **(2012)**	US	49	6610	7855	640 × 480 to 1024 × 52	6 × 6 to 167 × 168	All annotations
**UKOD** **(2012)**	UK	100+	43,509	1200 (synthetic)	648 × 480	24 × 24	No
**RTSD** **(2013)**	Russia	140	N/A	80,000+ (synthetic)	1280 × 720	30 × 30	No

**Table 3 sensors-19-02093-t003:** Colors based approaches for TSDR system.

Techniques	Paper	Segmentation Methods	Advantages	Sign Type	No. of Test Images	Test Image Type
**Color Thresholding Segmentation**	[37]	RGB color segmentation	Simple	Any color	2000	N/A
	[38]	RGB color segmentation with enhancement of color	Fast and high detection rate	Red, blue, yellow	135	Video data
**HSI/HSV Transform**	[40]	HSI thresholding with addition for white signs	Segments adversely illuminated signs	Any color	N/A	High-res
	[33]	HSI color-based segmentation	Simple and fast	Any color	N/A	N/A
	[41]	RGB to HSI transformation	Segments adversely illuminated signs	Any color	N/A	Low-res
	[42]	RGB to HSI transformation	N/A	Red	N/A	Low-res
	[43]	RGB to HSI transformation	N/A	Any color	3028	Low-res
	[44]	HSI color-based segmentation	Simple and high accuracy rate	Red, blue	N/A	Video data
	[45]	HSI color-based segmentation	Simple and real time application	Any color	632	High-res
**Region Growing**	[48]	Started with seed and expand to group pixels with similar affinity	N/A	N/A	N/A	N/A
	[47]			N/A	N/A	High-res
**Color Indexing**	[50]	Comparison of two any-color images is done by comparing their color histogram	Straightforward, fast method	Any color	N/A	Low-res
	[49]			Any color	N/A	N/A
**Dynamic Pixel Aggregation**	[52]	Dynamic threshold in pixel aggregation on HSV color space	Hue instability reduced	Any color	620	Low-res
**CIECAM97 Model**	[54]	RGB to CIE XYZ transformation, then to LCH space using CIECAM97 model	Invariant in different lighting conditions	Red, blue	N/A	N/A
**YCbCr Color Space**	[55]	RGB to YCbCr transformation then dynamic thresholding is performed in Cr component to extract red object	Simple and high accuracy	Red	193	N/A
	[56]		High accuracy less processing time	Any color	N/A	Low-res

**Table 4 sensors-19-02093-t004:** Shape-based methods for TSDR system.

Technique	Paper	Overall Process	Recognition Feature	Advantages	Sign Type	No. of Test Image	Test Image Type
**Hough Transform**	[77]	Each pixel of edge image votes for the object center at object boundary	N/A	Invariant to in-plane rotation and viewing angle	Octagon, square, triangle	45	Low-res
	[78]		AdaBoost	High accuracy	Any sign	N/A	Low-res
	[79]		N/A	Robustness to illumination, scale, pose, viewpoint change and even partial occlusion	Red (circular), blue (square)	500+	Low-res
	[80]		N/A	Reducing memory consumption and increasing utilization Hough-based SVM	Any sign	3000	High-res
	[81]		N/A	Robustness	Red (circular)	N/A	768 × 580
	[59]		Random Forest	Improve efficiency of K-d tree, random forest and SVM	Triangular and circular	14,763	752 × 480 px
	[82]		SIFT and SURF based MLP	Applying another state refinement	Red circular	N/A	Video data
**Similarity Detection**	[52]	Computes a region and sets binary samples for representing each traffic sign shape.	NN	Straight forward method	Any color	620	Low-res
**DTM**	[61]	Capturing object shape by template hierarchy.	RBF Network	Detects objects of arbitrary shape	Circular and triangular	1000	360 × 288 px
**Edge Detection Feature**	[63]	A set of connected curves is found which indicates the boundaries of objects within the image.	Geometric matching	Invariant in translation, rotation and scaling	Any color	1000	640 × 480
	[64]		Normalized cross correlation	Reliability and high accuracy in real time	Speed limit sign	N/A	320 × 240 px video data
	[65]		N/A	Improved accuracy by training negative sample	Red (circular)	3907	Low-res
	[66]		N/A	Invariant in noise and lighting	Triangle, circular	847	High-res
	[67]		CDT	Invariant in noise and illumination	Red, blue, yellow		
**Edges with Haar-like Features**	[69]	Sums three pixel intensities and calculates the difference of sums by Haar-like features	CDT	Smoother and noise invariant	Rectangular, any color		Video data
	[70]		SVM	Fast method	Circular, triangular upside-down, rectangle and diamond		640 × 480 px video data

**Table 5 sensors-19-02093-t005:** Sign tracking based on Kalman Filter approaches.

Technique	Paper	Advantages	Performance
**Kalman Filter**	[82]	For avoiding incorrect assignment, rule-based approach utilizing combined distance direction difference is used.	N/A
	[89]	Takes less time in tracking and verifying	Using 320 × 240 pixel images, takes 0.1 s to 0.2 s.
	[88]	Used stereo parameters to reduce the error of stereo measurement	N/A
**Advanced Kalman Filter**	[85]	Fast and advanced method, high detection and tracking rate	Using 400 × 300 pixel images, can process 3.26 frames per second.

**Table 6 sensors-19-02093-t006:** Examples of TSDR systems using a template matching method.

Ref	Detection Feature	Advantages	True Positive Rate	False Positive Rate	No. of Test Images	Overall Accuracy	Time
**[90]**	RGB to HSV then contrast stretching	Fast and straight forward method	N/A	N/A	N/A	<95%	N/A
**[91]**	N/A		N/A	N/A	100	90.9%	N/A

**Table 7 sensors-19-02093-t007:** Examples of TSDR systems using a decision tree method.

Ref	Detection Feature	Advantages	True Positive Rate	False Positive Rate	No. of Test Images	Overall Accuracy	Time	Dataset
**[94]**	HOG based SVM	Used GTSRB and ETH 80 dataset and compared	90.9%	N/A	12,569	90.46%	17.9 ms	GTSRB and ETH 80
**[95,96]**		Used Gaussian weighting in HOG to improve performance by 15%	90%	N/A	12,569	97.2%	17.9 ms	Own created
**[92]**	MSER based HOG	Eliminating hand labeled database, robust to various lighting and illumination	83.3%	0.85	640 × 480 px video data	87.72%	N/A	Own created
**[97]**	HOG	Remove false alarm up to 94%	N/A	N/A	12,569	92.7%	17.9 ms	Own created

**Table 8 sensors-19-02093-t008:** Examples of TSDR systems using a genetic algorithm.

Ref	Detection Feature	Advantages	True Positive Rate	False Positive Rate	No. of Test Images	Overall Accuracy	Time
**[98,99]**	Genetic Algorithm	Unaffected by illumination problem	N/A	N/A	Video data	N/A	N/A

**Table 9 sensors-19-02093-t009:** Examples of TSDR systems using an ANN method.

Ref	Detection Feature	Advantages	True Positive Rate	False Positive Rate	No. of Test Images	Overall Accuracy	Time	Dataset
**[56]**	YCbCr and normalized cross correlation	Robustness and adaptability	0.96	0.08	640 × 480 px video data	97.6%	0.2 s	Own created
**[101]**	N/A	Flexibility and high accuracy	N/A	N/A	N/A	98.52–99.46%	N/A	Own created
**[106]**	Adaptive shape analysis	Invariant in illumination	N/A	N/A	220	95.4%	0.6 s	Own created
**[107]**	NN	Robustness	N/A	N/A	467	N/A	N/A	Own created
**[108]**	Bimodal binarization and thresholding	Compared TM and NN elaborately	0.96	0.08	640 × 480 px video data	97.6%	0.2 s	Own created

**Table 10 sensors-19-02093-t010:** Examples of TSDR systems using a deep learning method.

Ref	Detection Feature	Advantages	True Positive Rate	False Positive Rate	No. of Test Images	Overall Accuracy	Time	Dataset
**[115]**	Object bounding box prediction	Predicting position and precise boundary simultaneously	>0.88 mPA	<3 pixels	3,719	91.95%	N/A	GTSDB
**[120]**	YCbCr model	High accuracy and speed	N/A	N/A	Video data	98.6%	N/A	Own created
**[111]**	Color space thresholding	Implementing detection and classification	90.2%	2.4%	20,000	95%	N/A	GTSRB
**[121]**	SVM	Robust against illumination changes	N/A	N/A	Video data	97.9%	N/A	Own created
**[117]**	Scanning window with a Haar cascade detector	Enhanced detection capability with good time performance	N/A	N/A	16,630	99.36%	N/A	GTSRB

**Table 11 sensors-19-02093-t011:** Examples of TSDR systems using an AdaBoost method.

Ref	Detection Feature	Advantages	True Positive Rate	False Positive Rate	No. of Test Images	Overall Accuracy	Time	Dataset
**[123]**	Sobel edge detection	Comparison of SVM and AdaBoost	N/A	0.25	N/A	92%	N/A	Own created
**[124]**	AdaBoost	Fast	N/A	N/A	200	>90%	50 ms	Own created
**[125]**	AdaBoost	Invariant in speed, illumination and viewing angle	92.47%	0%	350	94%	51.86 ms	Own created
**[126]**	AdaBoost and CHT	Real-time and robust system with efficient SLS detection and recognition	0.97	0.26	1850	94.5%	30–40 ms	Own created
**[127]**	Haar-like method	Reliability and accuracy	0.9	0.4	200	92.7%	50 ms	Own created

**Table 12 sensors-19-02093-t012:** Examples of TSDR systems using a SVM method.

Ref	Detection Feature	Advantages	True Positive Rate	False Positive Rate	No. of Test Images	Overall Accuracy	Time	Dataset
**[44]**	DtBs and SVM	Fast, high accuracy	N/A	N/A	Video data	92.3%	N/A	GRAM
**[55]**	Gabor Filter	Simple and high accuracy	N/A	N/A	58	93.1%	N/A	Own created
**[130]**	CIELab and Ramer–Douglas–Peucker algorithm	Illumination proof and high accuracy	N/A	N/A	405	97%	N/A	Own created
**[131]**	RGB to HSI then shape analysis	Less processing time	N/A	N/A		92.6%	Avg. 5.67 s	Own created
**[88]**	Hough transform	Reliability and accuracy	N/A	N/A	Video data	Avg. 92.3%	35 ms	Own created
**[132]**	RGB to HIS then shape localization	Reduce the memory space and time for testing new sample	N/A	N/A	N/A	95%	N/A	Own created
**[133]**	MSER	Invariant in illumination and lighting condition	0.97	0.85	43,509	89.2%	N/A	Own created
**[134]**	HSI and edge detection	Less processing time	N/A	N/A	Video data	N/A	N/A	Own created
**[135]**	RGB to HSI	Identify the optimal image attributes	0.867	0.12	650	86.7%	0.125 s	Own created
**[136]**	Edge Adaptive Gabor Filtering	Reliability and Robustness	85.93%	11.62%	387	95.8%.	3.5–5 ms	Own created

**Table 13 sensors-19-02093-t013:** Examples of TSDR systems using the other methods.

Ref	Method	Detection Feature	Advantages	True Positive Rate	False Positive Rate	No. of Test Images	Overall Accuracy	Time	Dataset
**[138]**	SIFT matching	N/A	Effective in recognizing low light and damaged signs	N/A	N/A	60	N/A	N/A	Own created
**[34]**	Fringe-adjusted joint Transform Correlation	Color Feature Extraction using Gabor Filter	Excellent discrimination ability between object and non-object	783	217	587	N/A	N/A	Own created
**[139]**	Principal Component Analysis	HSV, CIECAM97 and PCA	High accuracy rate	N/A	N/A	N/A	99.2%	2.5 s	Own created
**[140]**	Improved Fast Radial Symmetry and Pictogram Distribution Histogram based SVM	RGB to LaB color space then IFRS detection	High accuracy rate	N/A	N/A	300	96.93%	N/A	Own created
**[144]**	Infrastructures of vehicles	N/A	Eliminating possibility of false positive rate because of ID coding	N/A	N/A	Video data	N/A.	N/A	Own created
**[145]**	FCM and Content Based Image Recorder	Fuzzy c means (FCM)	Effective in real time application	N/A	N/A	Video data	<80%	N/A	Own created
**[141]**	Template matching and 3D reconstruction algorithm	N/A	Very effective in recognizing damaged or occulted road signs	In 3D, 54 out of 63	In 3D, 6 out of 63 and 3 signs were missing	4800	N/A	N/A	Own created
**[142]**	Low Rank Matrix Recovery (LRMR)	N/A	Fast computation and parallel execution	N/A	N/A	40,000	97.51%	>0.2	GTSRB
**[143]**	Karhunen–Loeve Transform and MLP	Oriented gradient maps	Invariant in illumination an different lighting condition	N/A	N/A	12,600	95.9%	0.0054 s/image	GTSRB
**[35]**	Self-Organizing Map	N/A	Fast and accurate	N/A	N/A	N/A	<99%	N/A	Own created

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
