# Peer review of "Vision-Based Traffic Sign Detection and Recognition Systems: Current Trends and Challenges"

_sensors, 2019, doi:10.3390/s19092093_

Round 1

Reviewer 1 Report

In this paper, the authors have analyzed the main direction of the research in the field of automatic TSDR and categorized the main approaches into particular sections to make the topics easy to understand and visualized the overall research for future directions. In general, this paper is well written and easy to follow. I would like to accept this paper if the following concerns are carefully addressed.

(1) Although this paper is well written and easy to follow, there are still some some typos/grammar errors. I would like the authors to carefully proofread this paper and correct all these typos.

(2) Although the authors have conducted thorough literature review, some very relevant references are still missing in the current version. For example, "Semisupervised feature analysis by mining correlations among multiple tasks" and "Semantic pooling for complex event analysis in untrimmed videos" also employ similar methods for their task. The authors should include this in the revision.

(3) The authors may want to report the running time of their system in the revision? This can be served as a demonstration that their system can be applied to real-world applications.

Based on the above comments, I would like to accept this paper with major revision.

Author Response

Dear Professor,

Please find the attachment for the point by ponit responses to the reviewer 1 comments and suggestions.

Thank you,

M A Hannan, PhD

Reviewer 2 Report

My comments are given in the attached file.

Author Response

Dear Professor,

Please find th attachment for the point by ponit responses to the reviewer 2 comments and suggestions. 

Thank you,

M A Hanann, PhD

Round 2

Reviewer 1 Report

I think the authors have addressed all of my concerns. I have no further concerns and would like to accept this paper as it is.